# How Is Rejection Sensitivity Linked to Non-Suicidal Self-Injury? Exploring Social Anxiety and Regulatory Emotional Self-Efficacy as Explanatory Processes in a Longitudinal Study of Chinese Adolescents

**DOI:** 10.3390/bs14100943

**Published:** 2024-10-14

**Authors:** Junyan Zhao, Anna Li, Kunlin Li, Fengqing Zhao

**Affiliations:** 1Psychological Counseling Center, Capital Normal University, Beijing 100048, China; zhaojy@cnu.edu.cn; 2Beijing Haidian Psychological Rehabilitation Hospital, Beijing 100191, China; xinlikangfu2021@163.com; 3School of Psychology, Capital Normal University, Beijing 100048, China; 2233502093@cnu.edu.cn; 4School of Education, Zhengzhou University, Zhengzhou 450001, China

**Keywords:** non-suicidal self-injury, rejection sensitivity, social anxiety, regulatory emotional self-efficacy, adolescents

## Abstract

Early adolescents are at high risk for non-suicidal self-injurious behavior (NSSI). Based on the Rejection Sensitivity Model, the Experiential Avoidance Model, and the Affect Regulation Model of Self-Injury, this study aimed to explain how rejection sensitivity was related to NSSI among adolescents by unraveling the mediating role of social anxiety and the moderating role of regulatory emotional self-efficacy (RESE) in this relationship. A three-wave longitudinal investigation with a three-month interval was conducted among 726 adolescents (M_age_ = 13.47, SD = 0.95; 46.0% girls) from a middle school in North China. The Cross-Lagged Panel Models (RI-CLPMs) were utilized to estimate the associations among the study variables. The results indicated that the incidence rates of NSSI in the three measurements among adolescents were 33.3%, 30.3%, and 24.1%, respectively. Adolescents’ rejection sensitivity and NSSI showed a declining trend over time. Furthermore, rejection sensitivity predicted NSSI through the longitudinal mediating effect of social anxiety. RESE played a protective role in adolescents’ NSSI, but its moderating effect was not significant. The findings increase our understanding of the association between rejection sensitivity and NSSI in adolescents, and they benefit educators in conducting targeted interventions through improving adolescents’ rejection sensitivity and social anxiety to reduce the risk of NSSI.

## 1. Introduction

Non-suicidal self-injury (NSSI) refers to the direct and deliberate destruction of one’s own body tissue in the absence of lethal intent, which commonly takes the form of cutting, scratching, burning, biting, and hitting one’s head against a hard object [1]. Adolescents are at high risk for self-injury, which usually begins between the ages of 12 and 14 and may continue for years [2]. The prevalence of NSSI in community samples of adolescents is remarkably high. Previous studies found that approximately 23% of adolescents reported deliberately injuring themselves at least once in their life, and almost 19% in the previous year [3]. It is associated with emotional problems and a wide range of mental disorders, such as anxiety disorder, dysthymic disorder, and eating disorders [4,5,6]. Additionally, NSSI can predict suicide attempts in adolescents and is a valid predictor of suicidal behavior [7]. Therefore, it is of great theoretical and practical importance to identify the predictors of NSSI in adolescents and their underlying mechanisms.

Rejection sensitivity (RS) refers to a tendency to anxiously expect, readily perceive, and overreact to rejection [8]. Based on the Rejection Sensitivity Model, adolescents with high rejection sensitivity will be more sensitive to rejection cues and tend to feel rejected and produce a series of cognitive and emotional reactions (e.g., anxiety, remorse, pain, and hostility), which may eventually lead to a series of maladaptive behaviors like social withdrawal, aggression, and self-injury [9]. A few cross-sectional studies have focused directly on the correlation between rejection sensitivity and NSSI [10,11], but there are still some limitations that need further attention. First, studies have not detected the longitudinal association between rejection sensitivity and NSSI, which limits the possibility to reveal causal relations. Moreover, the research is unclear about the possible mediating and moderating mechanisms underlying the association between rejection sensitivity and NSSI. Therefore, it is necessary to examine the effects of rejection sensitivity on NSSI in adolescents through longitudinal investigation and explore the possible roles of social anxiety and regulatory emotional self-efficacy in it, which may provide educators with evidence to design targeted interventions to protect adolescents from self-injury and help students with self-injurious behaviors.

### 1.1. Rejection Sensitivity and Non-Suicidal Self-Injury

Rejection sensitivity can increase the risk for self-injurious thoughts and behaviors. Based on the Rejection Sensitivity Model [9], adolescents high in rejection sensitivity are very alert to rejection cues. They can easily perceive rejection, which triggers a range of cognitive and emotional responses, such as anxiety, self-blame, pain, and hostility, and eventually lead to maladaptive behavioral responses, such as social withdrawal, aggression, and self-injury. Self-injury, on the one hand, can serve as a way to alleviate their negative emotions when they experience distress or pain when they confront interpersonal difficulties and challenges. On the other hand, self-injury and the scars from it can attract attention and care from others, serving as a strategy to avoid rejection or punishment [11].

The social–cognitive model of rejection sensitivity explains that the experience of rejection in childhood causes people to learn to associate rejection with certain situations and cues [12]. These cues will activate the anxious expectations of rejection, and these defensive expectations of rejection are key components of rejection sensitivity. When experiencing similar interpersonal situations in social settings, these anxious expectations or schema are very easily activated so that neutral or ambiguous interpersonal cues are perceived as signals of rejection, triggering the individual’s hostility and aggression. Such negative emotional and behavioral responses may be the underlying cause triggering real rejection by others. This real rejection by others reinforces their prior anxious expectations of rejection, thus creating a self-fulfilling prophecy. Therefore, individuals with high rejection sensitivity may experience more denial and social rejection because of their sensitivity to rejection.

Several empirical studies have attempted to identify the relationship between rejection sensitivity and self-injury [10,11,13,14,15], and most studies agree that rejection sensitivity is an important predictor of NSSI in adults [10,13] and adolescents [11,14,15]. These results suggest that a predisposition to rejection sensitivity may increase the risk of NSSI in general adolescents. In addition, a meta-analysis claims that rejection is associated with the risk of self-harm and suicidal ideation in adulthood [13]. Another meta-analysis indicates that the longitudinal correlation between rejection sensitivity and depression, anxiety, and loneliness is stable over time [16], which indirectly supports the positive association between rejection sensitivity and NSSI.

### 1.2. The Mediating Role of Social Anxiety

Social anxiety (SA) refers to negative emotional experiences such as anxiety, nervousness, and shyness experienced by people during social activities, which can lead to a certain degree of distress and impaired social functioning and exert negative effects on the individual’s social activities and interpersonal relationships, accompanied by avoidance behaviors [17]. Based on the Experiential Avoidance Model, self-injury is a way to escape or avoid internal unpleasant experiences, especially to avoid negative emotional experiences [18]. Social anxiety, as a typical negative emotion, may be a contributing factor to self-injury. On the one hand, individuals with social anxiety are not only afraid of negative evaluations but also prone to negative emotions when receiving positive evaluations [19]. Therefore, they are less likely to have positive affective experiences in their daily lives. On the other hand, social anxiety reduces the possibility to establish relationship with others, which may be detrimental in satisfying the basic needs for belongingness and connection. Previous studies indicate that social anxiety is significantly and positively associated with suicidal ideation [5,20,21]. In addition, compared to non-self-injurers, self-injurers showed higher levels of social anxiety and avoidance of peers [21].

As to the relation between rejection sensitivity and social anxiety, these two psychological constructs both involve perceived or actual anxiety in interpersonal threatening situations, but they show significant differences. Rejection sensitivity focuses on the anxieties and expectations of being rejected by others and emphasizes the individual’s expectations of others’ reactions, while social anxiety focuses on general self-presentation and emphasizes the individual’s reactions to social interactions. Previous studies indicate that the cognitive and affective bias involved in rejection sensitivity is a risk factor for social anxiety, and longitudinal studies show that the anxious anticipation of rejection is the only predictor of social anxiety and social avoidance among adolescents [22,23]. In addition, individuals with high rejection sensitivity use more expressive inhibition and less cognitive reappraisal when regulating their emotions, and thus have higher levels of social anxiety [24].

Based on the above theoretical and empirical studies, we suggest that social anxiety may serve as a mediator between rejection sensitivity and self-injury. In particular, those with high rejection sensitivity may experience more anxiety during social activities due to the fear of being rejected. Being chronically anxious may cause higher levels of social anxiety, and people with high levels of social anxiety may resort to non-adaptive behaviors such as self-injury to escape or alleviate this anxiety. If self-injurious behavior is negatively reinforced to relieve the individual’s anxiety, it is more likely that individuals will choose to relieve their anxiety by self-injury when they feel intense anxiety in social situations again.

### 1.3. The Moderating Role of Regulatory Emotional Self-Efficacy

Regulatory emotional self-efficacy (RESE) refers to an individual’s confidence in his or her ability to effectively regulate his or her own emotional states [25]. It consists of three main aspects: the ability to perceive emotional states, the ability to empathize with the feelings of others, and the ability to manage positive or negative emotions. RESE has previously been divided into two categories: perceived self-efficacy in expressing positive affect (POS) and perceived self-efficacy in managing negative affect (NEG). POS refers to the ability to feel and express positive events and emotions, and NEG refers to the ability to rationally cope and regulate in the face of frustration and negative emotions [25]. On this basis, Caprara et al. [26] subdivided NEG into two dimensions, perceived self-efficacy in managing anger/irritation (ANG) and perceived self-efficacy in managing despondency/distress (DES), thus constructing a second-order latent variable model. Wen et al. [27] conducted a factor analysis of regulatory emotional self-efficacy in a Chinese sample and obtained a first-order latent variable model that had three dimensions: POS, ANG, and DES.

RESE may be a protective factor for self-injurious behavior. Based on the Affect Regulation Model of Self-Injury, self-injury is an attempt to evoke, express, or manage strong negative emotions to achieve emotional regulation [28]. Individuals who have higher confidence to manage and regulate their emotions tend to use less maladaptive coping strategies such as self-injury. A study comparing adolescents who engaged in self-injury and those who never self-injured in Israel found that the former had significantly poorer RESE and the greater use of avoidance coping mechanisms [29]. Indirect evidence comes from the strong association between RESE and mental health. Previous studies show that a high sense of RESE can improve individuals’ positive expectations for the future and positive self-concept, so that individuals can feel more positive emotions and effectively improve their subjective well-being [30]. Meanwhile, a low sense of RESE is closely associated with individual internalizing problems, such as anxiety and depression, and externalizing problems such as aggression [31].

Furthermore, RESE may serve as a moderator in the association between social anxiety and NSSI. Previous studies have indicated that RESE has a significant moderating effect between negative emotions and self-injurious behaviors [32] and buffers the effect of emotional reactivity on NSSI [33]. This is because the level of RESE implies the confidence that individuals have in regulating their negative emotions. When adolescents experience intense negative emotions, if they hold the belief that they cannot regulate these negative emotions, it can diminish their ability to effectively manage them. Ultimately, this may lead to the use of maladaptive strategies, such as self-harm, to alleviate emotional distress [33].

### 1.4. The Current Study

Based on the abovementioned aspects, taking the Rejection Sensitivity Model, the Experiential Avoidance Model, and the Affect Regulation Model of Self-Injury as the framework, this study aimed to explore the effect of rejection sensitivity on NSSI among adolescents in China; test the mediating role of social anxiety between rejection sensitivity and NSSI; and examine the moderating role of RESE in the relationship among rejection sensitivity, social anxiety, and NSSI. The hypotheses for this study were as follows.

**Hypothesis 1.** 
*Rejection sensitivity at T1 would significantly predict NSSI at T1, T2, and T3 among adolescents.*


**Hypothesis 2.** 
*Social anxiety at T2 would mediate the relationship between rejection sensitivity at T1 and NSSI at T3 in adolescents.*


**Hypothesis 3.** 
*RESE at T3 would moderate the association between social anxiety at T2 and NSSI at T3 in adolescents.*


## 2. Materials and Methods

### 2.1. Participants

The participants were students from 12 classes within Grades 7, 8, and 9 from a key middle school in a county town in Hebei Province in China. Participants were invited and agreed to participate in the study. Three consecutive questionnaire assessments were administered at 3-month intervals. The first two assessments were conducted in September 2021 and December 2021. The last assessment was conducted one month later in April 2022 because of the delayed start of school due to COVID-19. The students were in the school at the time of the assessments. A total of 882 questionnaires were distributed in the first assessment, 841 in the second assessment, and 774 in the last assessment. Eliminating the participants who did not participate in all three investigations or reported high suicidal ideation on the Self-Harm Scale or high non-suicidal self-injury scores exceeding three standard deviations, 726 participants (female = 334, 46.01%; only child = 140, 19.30%; Grade 7 = 276, Grade 8 = 212, Grade 9 = 238) were considered valid. The age of the participants ranged from 11 to 16 (M = 13.47 years, SD = 0.95).

### 2.2. Measures

#### 2.2.1. Rejection Sensitivity

The Chinese version of the Children’s Rejection Sensitivity Questionnaire (CRSQ), developed by Downey et al. (1998), was used to test adolescents’ rejection sensitivity [34,35]. Six situations from the original 12 were selected as hypothetical situations and revised into the Chinese version of the CRSQ. Each of these consisted of 3 questions about children’s anxiety, anger, and rejection anticipation. Participants were asked to what extent they agreed with the statement on a 6-point scale; 1 represented the lowest level and 6 represented the highest level. Adolescents’ anxious expectation of rejection was generated for each of the 6 situations by multiplying the rejection expectation by the degree of anxiety over its occurrence for each situation, and then averaging this across all 6 situations. Similarly, adolescents’ anger expectation was generated by averaging across the 6 situations the product of rejection expectation and the degree of anger over it. A total score on RS can be computed by adding the scores for anxious and angry expectations of rejection and dividing it by two [36]. In this study, the Cronbach’s α values for the three measures of the questionnaire were 0.76, 0.81, and 0.81, respectively. In this study, the measure had good validity and the fit indices were as follows: χ^2^/df = 3.86, RMSEA = 0.06, CFI = 0.93, TLI = 0.90, and SRMR = 0.04.

#### 2.2.2. Social Anxiety

The Social Interaction Anxiety Scale (SIAS), developed by Mattick et al. (1998) and revised by Ye et al. (2007) [37,38], consists of 19 questions with a 5-point scale, where 1 = totally disagree and 5 = totally agree, with higher total scores indicating a higher level of social anxiety. In this study, the Cronbach’s α values for the three measurements of this questionnaire were 0.89, 0.91, and 0.92, respectively.

#### 2.2.3. Regulatory Emotional Self-Efficacy

The Regulatory Emotional Efficacy Scale (RESE) was developed by Caprara et al. (2008) and translated by Wen et al. (2009) [26,27]. The RESE consists of 3 dimensions, which are perceived POS, DES, and ANG. Each dimension consists of 4 questions, so there are 12 items in total. Each of these is scored on a 5-point scale, where 1 = totally disagree and 5 = totally agree. A higher total score means that the individual has a higher sense of RESE. Since positive emotions were not considered in this study, the combined scores of the DES and ANG dimensions were used to represent the subjects’ RESE. The reliability of this scale is good among junior high school students [39], and the Cronbach’s α values of the questionnaire in this study were 0.78, 0.86, and 0.87 for the three measures, respectively.

#### 2.2.4. Non-Suicidal Self-Injury

The Adolescent Self-Harm Scale assesses 18 self-injurious behaviors and one open-ended questions [40]. Participants reported the frequency and severity of each non-suicidal self-injury behavior in the previous three months, with the frequency scored on a 4-point scale (0 = 0 times, 1 = 1 time, 2 = 2 to 4 times, and 3 = 5 or more times) and severity on a 5-point scale (0 = “none” to 4 = “very severe”). The level of NSSI was generated by multiplying the score of the frequency and the score of the severity (the level of NSSI in the following refers to the product of frequency and severity). The Cronbach’s α values for the three measures of the questionnaire were 0.95, 0.95, and 0.96, respectively.

### 2.3. Procedure

First, ethical approval was obtained from the first author’s institution prior to conducting the investigation. Then, the researcher contacted the schools to introduce the purpose, content, and administration procedures of the study and obtained consent from the schools. Moreover, the trained head teacher in each class explained the purpose of the study and the notes for the completion of the questionnaires; they also obtained informed consent from the parents and assent from the students before they voluntarily completed the questionnaires. It took about 12 min to complete the whole set of questionnaires.

### 2.4. Data Analysis

First, descriptive statistics, correlations among variables, an attrition analysis, and common method biases were obtained using SPSS 22.0. Second, a repeated-measures analysis of variance was used to compare the levels of rejection sensitivity and NSSI at the three time points. Third, structural equation models (SEMs) were developed and analyzed in three steps using Mplus 8.3 according to the purposes of the study. In step one, an SEM was developed to examine the longitudinal effect of rejection sensitivity on NSSI. In step two, a second-order cross-lagged panel model (CLPM) was used to test the mediation effect of social anxiety between rejection sensitivity and NSSI. In step three, an SEM was used to establish the moderated mediation model to explore the moderating role of RESE in the association among rejection sensitivity at T1, social anxiety at T2, and non-suicidal self-injury at T3.

## 3. Results

### 3.1. Attrition Analysis

This study used an ANOVA to compare the effectively tracked sample and the lost sample in the four variables at T1. As is shown in Table 1, there was no significant difference (*p* > 0.05) between the two groups of subjects in terms of rejection sensitivity, social anxiety, and regulatory emotional self-efficacy. However, there was a marginally significant difference in the level of non-suicidal self-injury. This might be because the students whose scores in NSSI were above three standard deviations and reported suicidal thoughts at T1 were withdrawn from the experiment in consideration of the students’ physical and mental health.

### 3.2. Common Method Bias

In order to control for common method bias, this study used Harman’s single-factor test to conduct an exploratory factor analysis on all entries in the questionnaire, and the results showed that the variance explained by the first factor was 13.88%, 20.17%, and 22.30% in the three measurements, which were less than 40% of the judgmental criteria. This indicated that the data of the present study were not significantly affected by common method bias.

### 3.3. Preliminary Statistics

The distribution of the NSSI levels at the three time points indicated that the incidence rate was 33.3% at T1, 30.3% at T2, and 24.1% at T3. Overall, the incidence rate of NSSI gradually decreased across the three measures. A chi-square test showed that there was a significant difference between the incidence rates of NSSI at T1, T2, and T3 (*χ*^2^ = 15.53, *df* = 2, *p* < 0.001). The post hoc test showed that the non-suicidal self-injury incidence rate at T3 was significantly lower than that at T1 and T2 (*p* < 0.01), while the difference between T1 and T2 was not significant (*p* = 0.21). In addition, the percentage of middle school students who reported no NSSI (scored 0) at all three measurement time points was 51.9%, and the percentage of middle school students who had self-injurious behaviors once or more was 48.1%.

The results of the descriptive statistical analysis of the variables are presented in Table 2. The results indicated that both the concurrent and subsequent correlations between rejection sensitivity, social anxiety, RESE, and NSSI were significant for middle school students (|*r*| 0.10~0.72, *p* < 0.01).

Then, a repeated-measures ANOVA was conducted with the measurement time (T1, T2, and T3) as the within-subjects factor and rejection sensitivity as the dependent variable. The results revealed a discernible decline in rejection sensitivity over time, *F*_(2,724)_ = 24.22, *p* < 0.001, η_P_^2^ = 0.06. Post hoc analyses indicated that rejection sensitivity at T3 markedly decreased compared to T2 (*p* < 0.01), and rejection sensitivity at T2 significantly decreased compared to T1 (*p* < 0.001).

Furthermore, in order to reveal the changes in the NSSI levels and the possible effects of the grade, a 3 measurement time (T1, T2, and T3) × 3 grade (Grade 7, 8, and 9) repeated-measures ANOVA was conducted with the NSSI level as the dependent variable, the measurement time as a within-subjects factor, and the grade level as a between-subjects factor. The results showed a significant main effect of the measurement time (*F*_(2,723)_ = 8.13, *p* < 0.001, η_P_^2^ = 0.02). NSSI at T3 was significantly lower than at T2 (*p* < 0.001) and T1 (*p* < 0.05), while there was no significant difference between T2 and T1 (*p* = 0.10). The main effect of the grade was significant (*F*_(2,723)_ = 5.38, *p* < 0.01, η_P_^2^ = 0.02). Students in Grade 9 had significantly lower levels of NSSI than Grade 8 (*p* < 0.05) and Grade 7 (*p* < 0.01), and there was no significant difference between Grade 8 and Grade 7 (*p* = 0.98). The interaction effect of the measurement time and grade was significant, *F*_(4,1446)_ = 7.10, *p* < 0.001, η_P_^2^ = 0.02. A simple effect analysis revealed that, across the three time points, the difference in the NSSI levels of Grade 7 (*F*_(2,274)_ = 0.37, *p* = 0.69, η_P_^2^ = 0.00) and Grade 9 (*F*_(2,236)_ = 1.28, *p* = 0.28, η_P_^2^ = 0.01) was not significant, but it was significant for students in Grade 8, (*F*_(2,210)_ = 10.57, *p* < 0.001, η_P_^2^ = 0.09). The NSSI scores of students in Grade 7 showed a slight but not significant increase over time, and the NSSI in Grade 9 showed a slight but not significant decrease over time, whereas the NSSI of Grade 8 students showed a fluctuation, increasing from T1 to T2 and then decreasing from T2 to T3.

### 3.4. The Effect of Rejection Sensitivity on Non-Suicidal Self-Injury

To test the role of rejection sensitivity as a longitudinal predictor of non-suicidal self-injury in middle school students, a cross-lagged panel model was established using Mplus 8.3 (see Figure 1). The model showed a good fit as the fit indices were as follows: *χ*^2^ = 23.73, *df* = 8, *χ*^2^*/df* = 2.97, RMSEA = 0.05, CFI = 0.96, TLI = 0.92, and SRMR = 0.04. Rejection sensitivity at T1 was a significant positive predictor of NSSI at T1 (β = 0.20, *p* < 0.001, 90% CI [0.10, 0.26]), but could not predict NSSI at T2 (β = 0.06, *p* = 0.23, 90% CI [−0.10, 0.15]) and NSSI at T3 (β = 0.04, *p* = 0.35, 90% CI [−0.08, 0.13]). Rejection sensitivity at T2 could not significantly predict NSSI at T2 (β = 0.08, *p* = 0.67, 90% CI [−0.04, 0.16]) and NSSI at T3 (β = −0.01, *p* = 0.18, 90% CI [−0.13, 0.06]). Rejection sensitivity at T3 was a significant positive predictor of NSSI at T3 (β = 0.16, *p* < 0.001, 90% CI [0.05, 0.25]). In summary, rejection sensitivity could positively predict NSSI in middle school students concurrently at both T1 and T3. However, the results did not support the diachronic predictive role of rejection sensitivity for NSSI.

### 3.5. The Longitudinal Mediation Effect of Social Anxiety in Rejection Sensitivity and Non-Suicidal Self-Injury

A second-order lagged CLPM was constructed to test the longitudinal mediation effect of social anxiety in the association between rejection sensitivity and NSSI. The gender and grade were controlled in the analysis. The results are shown in Figure 2. The fit indices of this second-order lagged CLPM model were acceptable, *χ*^2^*/df* = 4.45, RMSEA = 0.07, CFI = 0.96, TLI = 0.93, and SRMR = 0.05. Rejection sensitivity at T1 positively predicted social anxiety at T2 (β = 0.15, *p* < 0.001, 90% CI [0.07, 0.20]) and social anxiety at T2 positively predicted NSSI at T3 (β = 0.09, *p* < 0.01, 90% CI [0.00, 0.14]). The mediation effect analysis revealed that the mediation effect of social anxiety at T2 between rejection sensitivity at T1 and NSSI at T3 was 0.02 (*p* < 0.05, 90% CI [0.00,0.03]). In addition, rejection sensitivity at T2 predicted social anxiety at T3 (β = 0.09, *p* < 0.01, 90% CI [0.00, 0.15]).

### 3.6. The Moderated Mediation Effect Analysis

Furthermore, the moderated mediation effect of RESE in the association among rejection sensitivity, social anxiety, and NSSI was tested. The results (Figure 3) showed that RESE at T3 was a negative predictor of NSSI at T3 (β = −0.18, *p* < 0.001, 90% CI [−0.27, −0.10]), but the interaction of RESE at T3 and social anxiety at T2 was not a significant predictor of NSSI at T3 (β = −0.03, *p* = 0.43, 90% CI [−0.11, 0.03]); thus, RESE did not significantly moderate the second half of the mediating pathway.

## 4. Discussion

This study followed 726 junior high school students for seven months with the aim of exploring the relationship among rejection sensitivity and NSSI and exploring the roles of social anxiety and RESE among junior high school students. The results indicated that the incidence of NSSI in the three measurements among middle students was 33.3%, 30.3%, and 24.1%, respectively, and the percentage of junior high school students who had self-injurious behaviors one or more times in a seven-month period was 48.1%. This is consistent with the results of previous studies [41], indicating that self-injurious behavior is quite common in Chinese adolescents and is a problem that needs to be taken seriously. Furthermore, the longitudinal effect of rejection sensitivity on NSSI was explained by adolescents’ social anxiety. Finally, RESE at T3 played a protective role in NSSI at T3, but it did not moderate the association between social anxiety at T2 and NSSI at T3.

### 4.1. The Mediating Role of Social Anxiety in Rejection Sensitivity and NSSI

The correlational analysis revealed a significant positive association between rejection sensitivity and NSSI among middle school students. However, the results from the longitudinal predictive model indicate that rejection sensitivity concurrently predicts NSSI at T1 and T3, but does not longitudinally predict NSSI among middle school students, thus failing to support Hypothesis 1. This finding aligns with previous research findings [10,16], suggesting that rejection sensitivity does not directly predict self-injurious behavior in middle school students over time. In fact, prior research has indicated that rejection sensitivity can indirectly increase an individual’s risk of self-injury by influencing their emotional, cognitive, and motivational processes [10,15]. Thus, in the further analysis, we were concerned with the mediating role of social anxiety.

The results reveal that social anxiety serves as a longitudinal mediator between rejection sensitivity and NSSI among middle school students, thus supporting Hypothesis 2. In particular, adolescents who exhibit high rejection sensitivity will experience more social anxiety during interactions with peers and teachers, which in turn increases the risk of NSSI. First, the results suggest that rejection sensitivity can longitudinally predict social anxiety, which is consistent with previous findings [22,42]. Due to inherent cognitive–emotional biases, individuals with high rejection sensitivity exhibit more negative perceptions in interpersonal situations, readily perceiving rejection and displaying exaggerated reactions. Such interpersonal interactions contribute to heightened anxiety and a fear of future social encounters, thereby escalating their levels of social anxiety. Acknowledgment and acceptance among peers form the foundation for middle school students to establish interpersonal relationships. However, individuals with high rejection sensitivity are excessively concerned with whether their peers accept or reject them. At times, ambiguous cues in social contexts may be misinterpreted as rejection signals, impeding further engagement and connection with others, potentially leading to relationship breakdowns and establishing a vicious cycle. These results also validate the rejection sensitivity model, i.e., adolescents with higher levels of rejection sensitivity tend to experience more negative emotions (e.g., social anxiety), which eventually lead to self-injury and other maladaptive behaviors. Second, social anxiety can longitudinally predict self-injury in middle school students, which is consistent with previous findings [5] and supports the Experiential Avoidance Model. In other words, adolescents with lower levels of social anxiety may be inclined to alleviate their psychological distress through interpersonal interactions, seeking companionship and support from their peers to establish intimacy, obtain social support, and reinforce a sense of belonging. In contrast, individuals with higher levels of social anxiety may find it challenging to rely on peer support when facing stressors, leading to feelings of loneliness and helplessness. This propensity could potentially result in the accumulation of negative emotions, with NSSI serving as a coping mechanism for adolescents to alleviate the discomfort associated with these negative emotional experiences.

### 4.2. The Role of Regulatory Emotional Self-Efficacy

This study investigated adolescents’ RESE at T3 and explored its moderating role in the association between social anxiety and NSSI. The results revealed that RESE failed to modulate the latter part of the intermediary process, failing to validate Hypothesis 3. However, the direct effect of RESE on NSSI at T3 was significant. This might be because RESE and social anxiety have a moderate correlation (*rs* = −0.43~−0.31), i.e., those with high social anxiety tend to have lower levels of RESE. Although RESE is a protective factor for NSSI, when social anxiety or other negative emotions are too strong and exceed the extent of adolescents’ emotional regulation abilities, their RESE is low and it is difficult to help them to effectively regulate the relationship between negative emotions and self-harm. In this case, these adolescents may not be able to cope effectively with emotional difficulties, which can lead to the exacerbation of their mental health problems. Another possible reason is that the lockdown during COVID-19 may have aggravated existing conflicts in families, leading to an increased need to regulate emotions [43]; thus, RESE at T3 may fluctuate due to differences in emotional states and emotional regulation experiences. RESE at T3 may not be representative of adolescents’ real RESE. Future studies may explore the moderating role of RESE using the average RESE at three time points or using other statistical methods to analyze it more accurately.

Additionally, this study discovered a direct effect of RESE on NSSI in middle school students, which is consistent with previous research findings [32]. Previous researchers have identified RESE as a positive individual resource with significant protective effects on middle school students at risk of suicide [44]. Adolescents with higher RESE can more effectively manage negative emotions and stress, employing suitable emotional regulation strategies to cope with unfavorable situations, rather than resorting to maladaptive methods such as self-harm to evade distress. Since adolescents are in a period when their emotions fluctuate greatly and they are easily influenced by the environment, they should be encouraged to seek comprehensive psychological support and assistance to enhance their RESE, learning to manage their emotions and stress more effectively to prevent self-injury. What is noteworthy is that we only found a concurrent effect of RESE on NSSI at T3, but did not find a diachronic effect; thus, the results cannot reveal their longitudinal correlation, and generalization should be performed with caution.

### 4.3. Limitations and Future Directions

This study has several limitations. First, the use of convenience sampling in a middle school from a county town constrains the generalizability of the research findings. Future studies should broaden the area and age range of participants and employ more representative samples to enhance the external validity. Second, the sample, collected during COVID-19, lacks representativeness. Considering that the three investigations were all performed in the same school, when students were not isolated at home and had everyday social interactions with their peers and teachers, they may not accurately reflect the relationship among rejection sensitivity and NSSI. However, since students experienced isolation in Spring 2022, the adolescents’ rejection sensitivity and NSSI may have been influenced by the social context. As the results show, these adolescents’ rejection sensitivity saw a discernible decline and their NSSI varied across the three measures. Although there is no direct evidence about the degree to which this was influenced by the isolation, previous studies have indicated that COVID-19 had negative consequences for adolescents’ psychosocial development and increased the risk of NSSI [45]. Therefore, the results should be interpreted with caution in terms of generalizing them to adolescents in other social contexts. Third, the measurement method in this study relied on questionnaire assessments, which might not comprehensively capture the spectrum of self-injury situations. Future research could integrate qualitative methods, such as structured or semi-structured interviews, to gain a more in-depth understanding of the prevalence of self-injury among middle school students and the contributing factors, maintenance, or cessation of self-injury. Lastly, questionnaire assessments rely on self-reports, potentially being influenced by recall bias and social desirability. Therefore, future studies could combine experimental approaches to employ various research methods, enhancing the internal validity of the research.

### 4.4. Recommendations for Mental Health Education

Early adolescence is a high-incidence period for individuals to engage in self-injury. Self-injury poses a threat to adolescents’ physical and mental health and is closely related to suicide. In this study, the most common means of self-injury among middle school students were poking wounds, scratching, and hitting instead of scalding and electrocution, which is consistent with the findings of previous studies [46]. This may be because methods like scratches and hitting can be better self-controlled to avoid excessive self-injury resulting in threats to life. Teachers and parents should pay more attention to whether middle school students have obvious wounds on their arms or other parts of the body, so that they can detect them as early as possible and provide timely assistance.

Specifically, at the school and educator level, it is important to pay close attention to middle school students with a history of NSSI and those who exhibit suicidal ideation, and to prevent it in the early stage. Considering that students with high levels of social anxiety and rejection sensitivity have a susceptibility to self-injury and may perform self-injurious behaviors when they experience stress in interpersonal relationships, parent–child relationships, and academic activities, it is necessary to give students guidance at an early stage. For instance, a mental health course should be arranged to help students to understand the causation of rejection sensitivity and social anxiety, teach them strategies to deal with these difficulties, encourage them to correctly recognize negative emotions and better perceive emotions, teach them emotional regulation strategies, and make them more confident in dealing with their negative emotions. In addition, timely interventions should be arranged to protect students’ lives. For instance, mental health programs aimed at increasing regulatory emotional self-efficacy [47] can effectively improve adolescents’ mental health.

At the parental level, it is important to properly understand self-injurious behavior in adolescents. Parents need to take this seriously, put themselves in their children’s shoes to understand the difficulties that their children are experiencing, understand their children’s feelings, and provide them with stable family support. After their children have achieved stable emotions, parents can offer solutions from their own experience for their children to use as a reference. In addition, most self-injury behaviors in adolescents will gradually decrease with age until they subside. Therefore, parents should not exaggerate the severity of self-injury when they find that their children engage in such behaviors, so as to avoid adding to their children’s mental stress. Parents need to stabilize their own emotions; only with a correct understanding of self-injury can they better provide help to their children.

Finally, at the community level, community involvement is encouraged to help to develop stronger resilience among adolescents. For example, the community can provide spaces where DBT skills could be implemented and invite professionals to conduct regular lectures about the physical and psychological characteristics of adolescents, as well as effective methods of family communication.

## 5. Conclusions

Within the framework of the Rejection Sensitivity Model, Experience Avoidance Model, and Affect Regulation Model, this study conducted a longitudinal investigation to explore the relationship between rejection sensitivity and NSSI and then investigated the mediating role of social anxiety and the moderating role of RESE. The results indicate that the incidence of NSSI in the three measurements among middle students was 33.3%, 30.3%, and 24.1%, respectively, and rejection sensitivity indirectly predicted NSSI longitudinally through social anxiety. In addition, the moderating effect of NESE was not significant, but it is a protective factor for adolescents’ NSSI.

## Figures and Tables

**Figure 1 behavsci-14-00943-f001:**
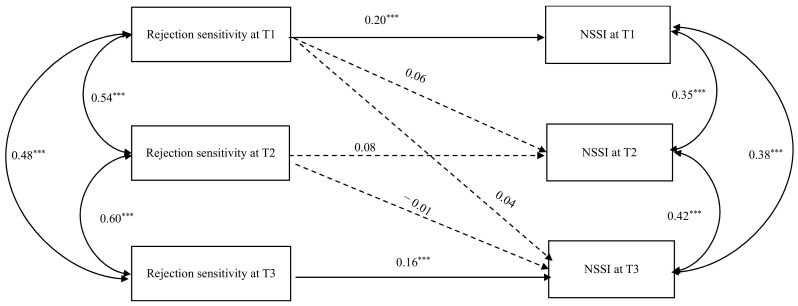
The association between rejection sensitivity and non-suicidal self-injury. *Note.* The coefficients listed in the figure are standardized coefficients. *** *p* < 0.001.

**Figure 2 behavsci-14-00943-f002:**
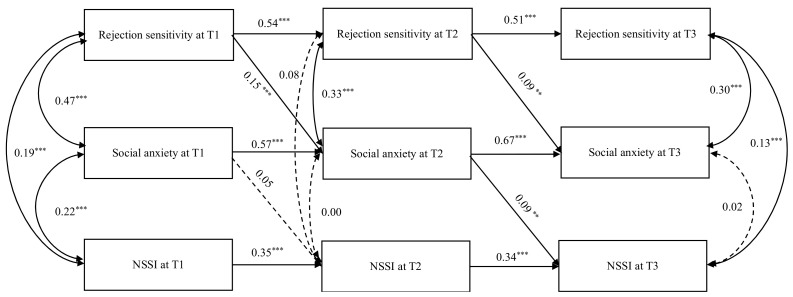
Longitudinal mediation effect of social anxiety in rejection sensitivity and NSSI. Notes. Covariates (gender and age) are not pictured for model simplicity and clarity. ** *p* < 0.01.*** *p* < 0.001.

**Figure 3 behavsci-14-00943-f003:**
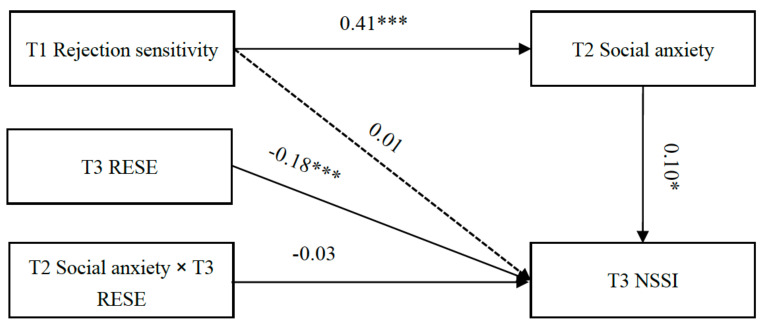
The moderated mediation analysis. * *p* < 0.05.*** *p* < 0.001.

**Table 1 behavsci-14-00943-t001:** Attrition analysis of the tracked sample and the lost sample.

Variable	Tracing Sample *n* = 774	Attritional Sample *n* = 108	*F*	*p*
RS	10.04 ± 4.57	10.61 ± 5.04	1.43	0.23
SA	53.04 ± 14.84	52.47 ± 15.25	0.14	0.71
RESE	24.80 ± 6.25	24.27 ± 6.64	0.69	0.41
NSSI	3.59 ± 9.79	5.78 ± 13.84	4.21	0.05

Notes. RS, rejection sensitivity; SA, social anxiety; RESE, regulatory emotional self-efficacy; NSSI, non-suicidal self-injury.

**Table 2 behavsci-14-00943-t002:** Descriptive statistics and correlations for study variables.

Variables	1	2	3	4	5	6	7	8	9	10
1. RS at T1	1									
2. RS at T2	0.54 ***	1								
3. RS at T3	0.48 ***	0.60 ***	1							
4. SA at T1	0.47 ***	0.36 ***	0.37 ***	1						
5. SA at T2	0.41 ***	0.49 ***	0.42 ***	0.67 ***	1					
6. SA at T3	0.36 ***	0.42 ***	0.51 ***	0.63 ***	0.72 ***	1				
7. RESE at T1	−0.41 ***	−0.27 ***	−0.23 ***	−0.38 ***	−0.34 ***	−0.33 ***	1			
8. RESE at T2	−0.40 ***	−0.44 ***	−0.35 ***	−0.36 ***	−0.43 ***	−0.40 ***	0.59 ***	1		
9. RESE at T3	−0.32 ***	−0.31 ***	−0.37 ***	−0.31 ***	−0.35 ***	−0.42 ***	0.48 ***	0.60 ***	1	
10. NSSI at T3	0.11 **	0.12 **	0.23 ***	0.12 ***	0.16 ***	0.16 ***	−0.12 ***	−0.11 ***	−0.21 ***	1
M	9.67	9	8.63	52.27	50.61	50.93	25.09	25.17	25.29	1.78
SD	4.08	4	3.82	14.29	14.43	14.48	6.07	6.8	6.76	4.75

** *p* < 0.01; *** *p* < 0.001.

## Data Availability

The data that support the findings of this study are contained within the article and are available from the corresponding author upon reasonable request.

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
