# Peer review of "How Is Rejection Sensitivity Linked to Non-Suicidal Self-Injury? Exploring Social Anxiety and Regulatory Emotional Self-Efficacy as Explanatory Processes in a Longitudinal Study of Chinese Adolescents"

_behavsci, 2024, doi:10.3390/bs14100943_

Round 1

Reviewer 1 Report

Comments and Suggestions for Authors

It would be important to review the follow-up information regarding the aim of the research since it is presented almost halfway through the introduction.

In point 3.5 of the results, a diachronic verification of the hypothesis is not achieved, which is not discussed in the corresponding section.

It is striking that the possible weight of the isolation period due to covid-19 is not considered relevant, although the study includes evaluations during that period.

The recommendations are appropriate to the findings.

Reviewer 2 Report

Comments and Suggestions for Authors

The paper is not only exciting but also essential for the field. I appreciate the effort and thought you've put into it.

I suggest defining rejection sensitivity when it is first mentioned instead of waiting until the third paragraph of the paper. Also, please mention the causations( very often seen in ADHD) of rejection sensitivity. 

The sentence on page 2, lines 50-51, is missing either an article or mechanism should be plural. Check sentences for grammar.

Grammatical error on page 2, lines 74-75. I recommend reviewing this part of the paper and making the necessary corrections.

Confusing sentence on page 2, lines 93-95. The last part of the sentence is without value.

The potential impact of the rural setting as a confounding factor in your study is not just intriguing, but also a promising avenue for further exploration. I encourage you to delve deeper into this connection, as it could significantly enhance the depth of your research and inspire new directions in the field.

Suggests that Figure 1 illustrates RESE as more of a protective factor rather than a contributing arrow.

In Discussion, authors could discuss community involvement in developing stronger resilience among adolescents, for example, spaces where DBT skills could be implemented.

Overall, the paper presents an interesting topic.

Comments on the Quality of English Language

 Minor grammatical mistakes found.

Reviewer 3 Report

Comments and Suggestions for Authors

Dears Authors

General comments

This is a very interesting subject and one that is in great need of study. I congratulate you on it.

I understand the purpose of the study “explain how rejection sensitivity was related to NSSI among rural adolescents by unraveling the mediating role of social anxiety and the moderating role of regulatory emotional self-efficacy (RESE) in this relationship”, However, I'd like it to be clearer why it's longitudinal. What did you expect to find different in 8 months? And why? What did you work on to promote results? Was there integration into a program? Was there a control group? You don't know the circumstances of the subject's life. What could have changed? And if there have been changes, what are they due to if none of the components have been worked on? Just the effect of the passage of time? Growth? I think you should reflect on this question.

By circumventing this issue and reformulating these premises, I think the basic results of your study could be interesting for the scientific community.

I present the comments for your reflection and possible consideration.

Title

- “How is rejection sensitivity linked to non-suicidal self-injury? Exploring social anxiety and regulatory emotional self-efficacy as explanatory processes in a longitudinal study of Chinese rural adolescents” - Clear, but it should be shorter and more directive - from a high school in the X region.

 Abstract

- Mage?

- It is not clear what intervention was carried out over the 6 months - it will be important to include. 6 or 8 months?

- You say “The findings broaden our understanding of the association between rejection sensitivity and NSSI in adolescents, and will benefit educators in identifying adolescents with high risk of NSSI and guiding intervention strategies.”- How do you do it?

- With an average age of 13.47 (between 11 and 16), can we call them teenagers?

- Keywords: why “adolescentes” – see previous comment

 1. Introduction

- Line 47- “A Few cross-sectional studies” - Uppercase typo

- Lines 47 and 48- You say “A Few cross-sectional studies have focused directly on the correlation between rejection sensitivity and NSSI”- add some fonts

- Lines 70-71- “Several empirical studies have attempted to identify the relationship between rejection sensitivity and self-injury,…”- What are they? Give some examples

- Lines 91-93- You say “Previous studies indicate that social anxiety is significantly and positively associated with suicidal ideation [19].”- mention several but only identify one

- Lines 103-104 - You say “and longitudinal studies show that anxious anticipation of rejection is the only 103 predictor of social anxiety and social avoidance among adolescents [21].” – once again, you refer to several studies but only identify one

- Lines 125-126- remove (Bandura et al., 2003)

- Line 126- “Caprara et al. (2008) and translated by Wen et al. (2009) [24, 25].”- or you should put it in instead: Caprara et al. [24] and translated by Wen et al. [25].

- Lines 129-131- “Wen et al. (2009) conducted a factor analysis of regulatory emotional self-efficacy in a Chinese sample and came up with a first-order latent variable model which has three dimensions: POS, ANG, and DES [25].” Or Wen et al. [25] conducted a factor analysis of regulatory emotional self-efficacy in a Chinese sample and came up with a first-order latent variable model which has three dimensions: POS, ANG, and DES.

 2. Materials and Methods

2.2. Measures

2.2.1. Rejection sensitivity

- Line 193- “All the items were answered on a 6-point”- what are they?

- Line 205- “consists of 19 questions with a 5-point scale,…” - what are they?

- Line 112- “items in total each of which is scored on a 5-point scale.” - what are they?

 2.1. Participants

- Lines 176-177- “Three consecutive questionnaire assessments were administered at 3- month interval. The first two assessments were conducted in September 2021 and December 2021.”- lack of information? 3-?

 2.3. Procedure

- Has the study been approved by an ethics committee? If so, you should include that information here.

- In my opinion, you should explain the application times here. “Three consecutive questionnaire assessments were administered at 3- month interval. The first two assessments were conducted in September 2021 and December 2021. The last assessment was done in April 2022 because of the delayed start of school due to the COVID-19.” And not in the participants

 4. Discussion

- Line 362- You say that “This study followed 726 junior high school students for six months with the aim of”- but if it was from September 2021 to April 2022, that's eight months, not six

- Lines 362-364- You mention with the aim of exploring the relationship among rejection sensitivity, social anxiety, RESE, and NSSI among junior high school students.”- or as you say in the lines 156-160- “this study aimed to explore the effect of rejection sensitivity on NSSI among adolescents in rural China, test the mediating role of social anxiety between rejection sensitivity and NSSI, and examine the moderating role of RESE in the relationship among rejection sensitivity, social anxiety, and NSSI.”- should homogenize

- Lines 364-367- You say that Results indicated that the incidence of NSSI in the 364 three measurements among middle students was 33.3%, 30.3%, and 24.1%, respectively, 365 and the percentage of junior high school students who had self-injurious behaviors one or 366 more times in a six-month period was 48.9%.”- six months? Review typo

- Lines 379-380- You mention as a relatively stable personality trait, does not directly predict self-injurious behavior in middle school students over time”- I have reservations about the way it is written - do children and young people aged between 11 and 16 show stable personality trait? You should support your assertion.

- Lines 382-384- When you say “The results reveal that social anxiety serves as a longitudinal mediator between rejection sensitivity and NSSI among middle school students, thus supporting Hypothesis 2.”- is confusing and unsupported. In my opinion, it should be reformulated and contextualized.

- Line 396- “These results also validate the rejection sensitivity model.”- explain more about how your concrete results validate this.

- Lines 406-408- You mention “Therefore, one effective approach to prevent adolescent NSSI is to disrupt the connection between negative emotions and self-harming actions.”- From my perspective, this conclusion lacks empirical support, based on your study.

- Lines 409-428- Confusing. You should review it, discussing your results in the literature and not moving on to other considerations that are already known.

- Lines 425-428- When you say “The prevention or intervention program for adolescent self-injury can start from improving adolescents’ self-efficacy in managing negative emotions, and reduce the occurrence of self-injury by enhancing their confidence in regulating their emotions.” How do you concretely support this in the results of your study?

- Lines 434-436- You indicate as one of the limitations “Secondly, the study primarily focused on exploring rejection sensitivity as a relatively stable personality trait and its impact on self-injury through internal cognitive and emotional mechanisms. It overlooked the influence of environmental factors."- I ask you to reflect on the ‘stable personality trait’ and I think that you must necessarily reflect, in addition to the environmental elements, on the COVI-19 pandemic in your results.

- In the discussion as a whole, there are several conclusions that are not supported by your results and that I think need to be revised.

 4.4. Recommendations for mental health education

- Lines 448- 482- very interesting information to share in other contexts. I don't think they add value to an original article. They are not derived from the results of your study.

- Identifying this need through incidence rates of NSSI (M= 29,23%- 33.3%, 30.3%, and 24.1%) reiterates the need for investment. You mention that “The mental health course should be arranged to help students to correctly recognize emotions, better perceive emotions, teach students the emotion regulation strategies, and make them more confident in dealing with their negative emotions” , which in my view is important to mention, however, it should be a suggestion that stems from the results of your study, that is, working on the issues of correctly recognizing emotions as the basic objective, but working directly on the elaboration of a project that promotes rejection sensitivity, social anxiety, and regulatory emotional self-efficacy would be important to share. Explain how according to the results of your study.

Thank you for the opportunity to read and comment on your study.

I hope I have contributed to a reflection that can further improve your work.

I wish you good work.
